# ARID1A promotes genomic stability through protecting telomere cohesion

Bo Zhao[1], Jianhuang Lin [1], Lijie Rong[2], Shuai Wu[1], Zhong Deng[1], Nail Fatkhutdinov[1], Joseph Zundell[1], Takeshi Fukumoto[1], Qin Liu [3], Andrew Kossenkov[4], Stephanie Jean[5], Mark G. Cadungog[5], Mark E. Borowsky[5], Ronny Drapkin[6], Paul M. Lieberman[1], Cory T. Abate-Shen[2] & Rugang Zhang [1]

ARID1A inactivation causes mitotic defects. Paradoxically, cancers with high *ARID1A* mutation rates typically lack copy number alterations (CNAs). Here, we show that ARID1A inactivation causes defects in telomere cohesion, which selectively eliminates gross chromosome aberrations during mitosis. ARID1A promotes the expression of cohesin subunit STAG1 that is specifically required for telomere cohesion. ARID1A inactivation causes telomere damage that can be rescued by STAG1 expression. Colony formation capability of single cells in $G_2/M$, but not $G_1$ phase, is significantly reduced by ARID1A inactivation. This correlates with an increase in apoptosis and a reduction in tumor growth. Compared with *ARID1A* wild-type tumors, *ARID1A*-mutated tumors display significantly less CNAs across multiple cancer types. Together, these results show that ARID1A inactivation is selective against gross chromosome aberrations through causing defects in telomere cohesion, which reconciles the long-standing paradox between the role of ARID1A in maintaining mitotic integrity and the lack of genomic instability in *ARID1A*-mutated cancers.

[1] Gene Expression and Regulation Program, The Wistar Institute, Philadelphia, PA 19104, USA. [2] Department of Pharmacology, Herbert Irving Comprehensive Cancer Center, Columbia University Irving Medical Center, New York, NY 10032, USA. [3] Molecular and Cellular Oncogenesis Program, The Wistar Institute, Philadelphia, PA 19104, USA. [4] Center for Systems and Computational Biology, The Wistar Institute, Philadelphia, PA 19104, USA. [5] Helen F. Graham Cancer Center & Research Institute, Newark, DE 19713, USA. [6] Department of Obstetrics and Gynecology, Perelman School of Medicine, University of Pennsylvania, Philadelphia, PA 19104, USA. Correspondence and requests for materials should be addressed to R.Z. (email: rzhang@wistar.org)

ARID1A, encoding a subunit of the BAF (mammalian SWI/SNF) complex, is among the genes that are most frequently mutated in human cancers[1,2]. For example, ARID1A is mutated in up to 60% of ovarian clear cell carcinomas (OCCCs)[3–5]. ARID1A functions as a tumor suppressor in OCCCs. Over 90% of ARID1A mutations in OCCCs are either frame-shift or nonsense, which leads to loss of ARID1A protein expression[3–5]. The ARID1A containing BAF complex remodels chromatin structure in an ATP dependent manner to modulate a number of processes that require DNA access such as transcription, DNA damage repair and replication[6]. In addition, ARID1A interacts with topoisomerase IIa (TOP2A) that resolves sister chromatids linked by catenated DNA strands during mitosis[7]. ARID1A is required for TOP2A's chromatin association and decatenation of newly replicated sister chromatids during mitosis[7]. Indeed, ARID1A inactivation leads to activation of the decatenation checkpoint and polyploidy in vitro[7,8]. These functions of ARID1A would predict large-scale genomic alterations and aneuploidy in ARID1A-mutated cancers caused by mitotic defects. Paradoxically, cancer types associated with high frequency of ARID1A mutations typically lack widespread genomic instability as measured by copy number alterations (CNA). For example, compared with high-grade serous ovarian cancer that is characterized by genomic instability and aneuploidy, OCCCs show relatively few large-scale CNA such as amplifications or deletions[5,9]. The molecular mechanism underlying this paradox remains to be elucidated.

Cohesin is a four subunit complex that is required for sister chromatid cohesion[10]. Sister chromatid cohesion is essential for accurate chromosome segregation and therefore cohesin is critical for genomic stability. In mammalian cells, cohesin consists of common SMC1, SMC3, and SCC1 subunits, and one of two mutually exclusive stromal antigen 1 (STAG1) or STAG2 subunits[10]. STAG1 mediates sister chromatid cohesion at telomeres, whereas STAG2 is required for sister chromatid cohesion at centromeres[11]. Indeed, STAG1 inactivation causes defects in telomere cohesion and chromosome mis-segregation during mitosis[11,12].

Here, we show that ARID1A inactivation causes defective telomere cohesion due to downregulation of STAG1, which acts selectively against genomic instability during mitosis. ARID1A promotes STAG1 expression. ARID1A inactivation causes telomere damage that can be rescued by STAG1 expression. Colony formation capability of single cells in $G_2/M$, but not $G_1$ phase, is significantly reduced by ARID1A inactivation. This correlates with an increase in apoptosis and a reduction in tumor growth. Compared with ARID1A wild-type tumors, ARID1A-mutated tumors display significantly less genomic instability as measured by CNA across multiple cancer types. Together, these results show that ARID1A inactivation is selective against gross chromosome aberrations through causing defects in telomere cohesion.

## Results

### ARID1A inactivation causes defective telomere cohesion.
When examining chromosome spreads in prometaphase mitotic shake-off cells, we discovered that compared with ARID1A wild-type OCCC RMG1 parental controls, isogenic ARID1A knockout (KO) RMG1 cells displayed a significant increase in the distance between distal ends of sister chromatids (Fig. 1a, b). Likewise, we observed an increase in the distance between distal ends of sister chromatids in chromosome spread of cells enriched by colcemid treatment (Fig. 1c, d). Similar observations were also made in ARID1A wild-type parental and the isogenic ARID1A KO OCCC OVCA429 cells (Supplementary Fig. 1a). Indeed, in a panel of

OCCC cell lines and primary cultures, compared with ARID1A wild-type OCCC cells, the distance between distal ends of sister chromatids in chromosome spread was significantly increased in ARID1A-mutated OCCC cells (Fig. 1e).

We next determined the telomere status using telomere fluorescence in situ hybridization (FISH) analysis. ARID1A KO correlated with an increase in loss of telomeric FISH signal in both RMG1 and OVCA429 ARID1A wild-type cells (Fig. 1f, g and Supplementary Fig. 1b). In addition, telomere signal loss was significantly greater in ARID1A-mutated cells compared with ARID1A wild-type cells in a panel of OCCC cell lines and primary cultures (Fig. 1h). Finally, compared with normal ovarian surface epithelial cells, telomere signal loss was significantly greater in OCCCs developed from a conditional Arid1a inactivation and Pik3ca activation mouse model (Fig. 1i, j). Notably, ARID1A inactivation did not decrease overall telomere length (Supplementary Fig. 1c). This suggests that observed telomere signal loss was not due to global reduction in telomere length.

Since telomere defects are known to induce DNA damage signaling and lead to telomere dysfunction-induced foci (TIFs)[13], we first examined the time course of expression of γH2AX in parental and ARID1A KO RMG1 and OVCA429 cells synchronized with double thymidine and followed by release (Fig. 2a and Supplementary Fig. 2a). Notably, compared with parental controls, ARID1A KO correlated with an increase in γH2AX expression and particularly in mitotic cells as indicated by phosphor-serine 10 histone H3 (pH3S10) expression (Fig. 2b). This correlated with an increase in γH2AX foci formation in metaphase cells upon ARID1A KO in both RMG1 and OVCA429 cells (Supplementary Fig. 2b–e). Consistently, γH2AX foci was significantly higher in metaphases of ARID1A-mutated compared with ARID1A wild-type cells in a panel of OCCC cell lines and primary cultures (Supplementary Fig. 2f). We next examined TIFs by directly assaying the co-localization of γH2AX and telomere in metaphase cells as confirmed by positive serine 10 phosphorylated histone H3 (H3 S10P) staining (Supplementary Fig. 2g, h). Indeed, there was a significant increase in γH2AX foci co-localized with telomere upon ARID1A KO in both RMG1 and OVCA429 cells (Fig. 2c, d and Supplementary Fig. 2i, j). Telomeres co-localized γH2AX foci were significantly higher in metaphases of ARID1A-mutated compared with ARID1A wild-type cells in panel of OCCC cell lines and primary cultures (Fig. 2e, f). Similar observations were also made in mouse bladder organoid cultures derived from wild-type control and P53[f/f]; Pten[f/f] with or without ARID1A KO mice (Fig. 2g, h), indicating this is not a tissue specific effect. As a control, ARID1A KO did not increase the co-localization of γH2AX with centromeres (Supplementary Fig. 2k, l). Conversely, re-expressing wild-type ARID1A in ARID1A mutant OVTOKO cells suppressed DNA damage at telomeres and reduced percentage of cells with anaphase bridge (Supplementary Fig. 2m–p). Together, these findings support the notion that ARID1A inactivation causes telomere cohesion defects, telomere signal loss and DNA damage signaling at telomeres.

### ARID1A inactivation increases chromosomal defects.
Telomere defects are known to induce chromosomal defects during mitosis[13]. Consistent with previous reports[7,8], we observed an increase in chromosomal defects such as anaphase bridge and lagging chromosomes during mitosis in ARID1A KO RMG1 and OVCA429 cells compared with parental controls (Fig. 3a–c and Supplementary Fig. 3a, b). Notably, the anaphase bridges and lagging chromosomes observed in ARID1A KO cells were positive for telomere protein TRF1, indicating that they were originated from the telomeres (Fig. 3d). Likewise, compared with

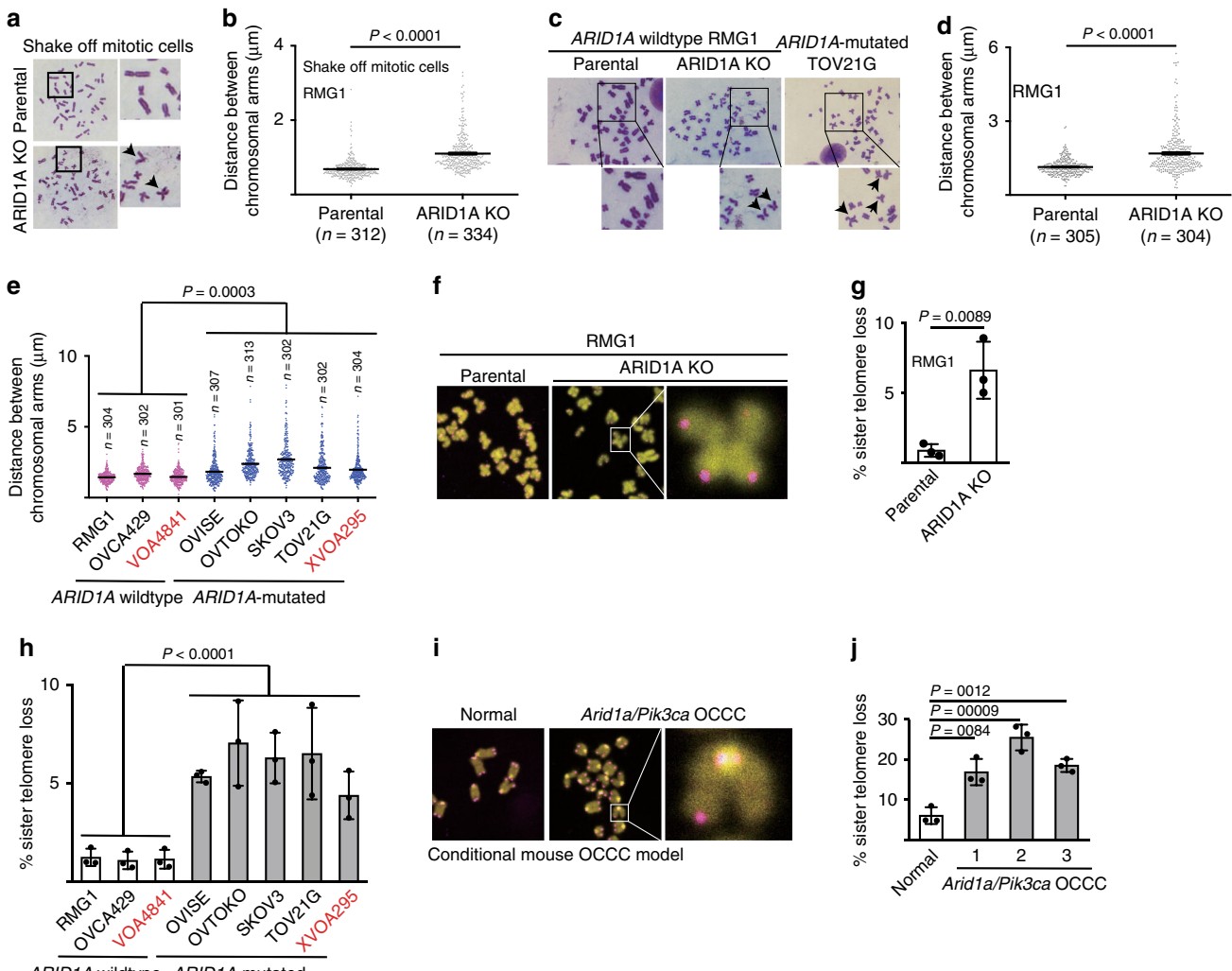

**Fig. 1** ARID1A inactivation causes defective telomere cohesion. **a, b** Representative images of prometaphase chromosome spreads (**a**) and quantification of distance between distal ends of sister chromatids (**b**) enriched by mitotic shake-off from parental and *ARID1A* knockout RMG1 cells. **c–e** Representative images of chromosome spreads (**c**) and quantification of distance between distal ends of sister chromatids (**d**) enriched by colcemid treatment from parental and *ARID1A* knockout RMG1 cells, and *ARID1A* mutated TOV21G cells. And quantification of distance between distal ends of sister chromatids enriched by colcemid treatment from the indicated clear cell ovarian cancer cell lines or primary cultures highlighted in red (**e**). **f, g** Representative images of telomere fluorescent in situ hybridization (**f**) and quantification of mitotic telomere signal loss (**g**) in parental and *ARID1A* knockout RMG1 cells. **h** Quantification of mitotic telomere signal loss in the indicated clear cell ovarian cancer cell lines. **i, j** Representative images of telomere fluorescent in situ hybridization (**i**) and quantification of mitotic telomere signal loss (**j**) in cells isolated from normal mouse ovary and $Arid1a^{-/-}/Pik3ca^{H1047R}$ genetic clear cell ovarian tumors respectively. $n = 3$ independent experiments unless otherwise stated. Data represent mean ± s.e.m. $P$ values were calculated using a two-tailed $t$ test except in 1e and 1h by multilevel mixed-effects models

*ARID1A* wild-type cells, *ARID1A*-mutated cells displayed a significantly higher percentage of cells with anaphase bridges and lagging chromosomes (Fig. 3e, f). Similar observations were also made using human OCCC patient-derived xenografts (PDXs) based on H&E staining (Fig. 3g, h and Supplementary Fig. 3c). Another consequence of telomere loss or uncapping is chromosomal fusion[13]. Indeed, ARID1A KO increased the percentage of cells with chromosomal fusion in both RMG1 and OVCA429 *ARID1A* wild-type cells (Fig. 3i, j and Supplementary Fig. 3d). In addition, compared with *ARID1A* wild-type cells, *ARID1A*-mutated cells displayed a significantly higher percentage of cells with chromosomal fusion (Fig. 3k). Finally, live cell imaging showed that compared with *ARID1A* wild-type cells, ARID1A KO or mutant cells displayed chromosomal defects such as lagging and chromosomal bridges during mitosis (Fig. 3l and Supplementary Movies 1–3). Consistent with previous reports[8], we

also observed an increase in mitosis duration in ARID1A KO cells compared with parental controls (Fig. 3m).

**ARID1A directly promotes STAG1 expression**. To determine the mechanism underlying the observed defects in telomere cohesion, we cross-referenced ARID1A chromatin immunoprecipitation (ChIP) followed by next generation sequencing (ChIP-seq) with RNA sequencing datasets in parental and ARID1A KO RMG1 cells[14] (Supplementary Fig. 4a). We focused on genes that are implicated in the functionality of chromosome segregation that includes sister chromatid cohesion. Notably, the STAG1 subunit of the cohesin complex is a direct target of ARID1A that is downregulated in ARID1A KO compared with control cells (Fig. 4a and Supplementary Fig. 4a). We characterized the downregulation of STAG1 by ARID1A

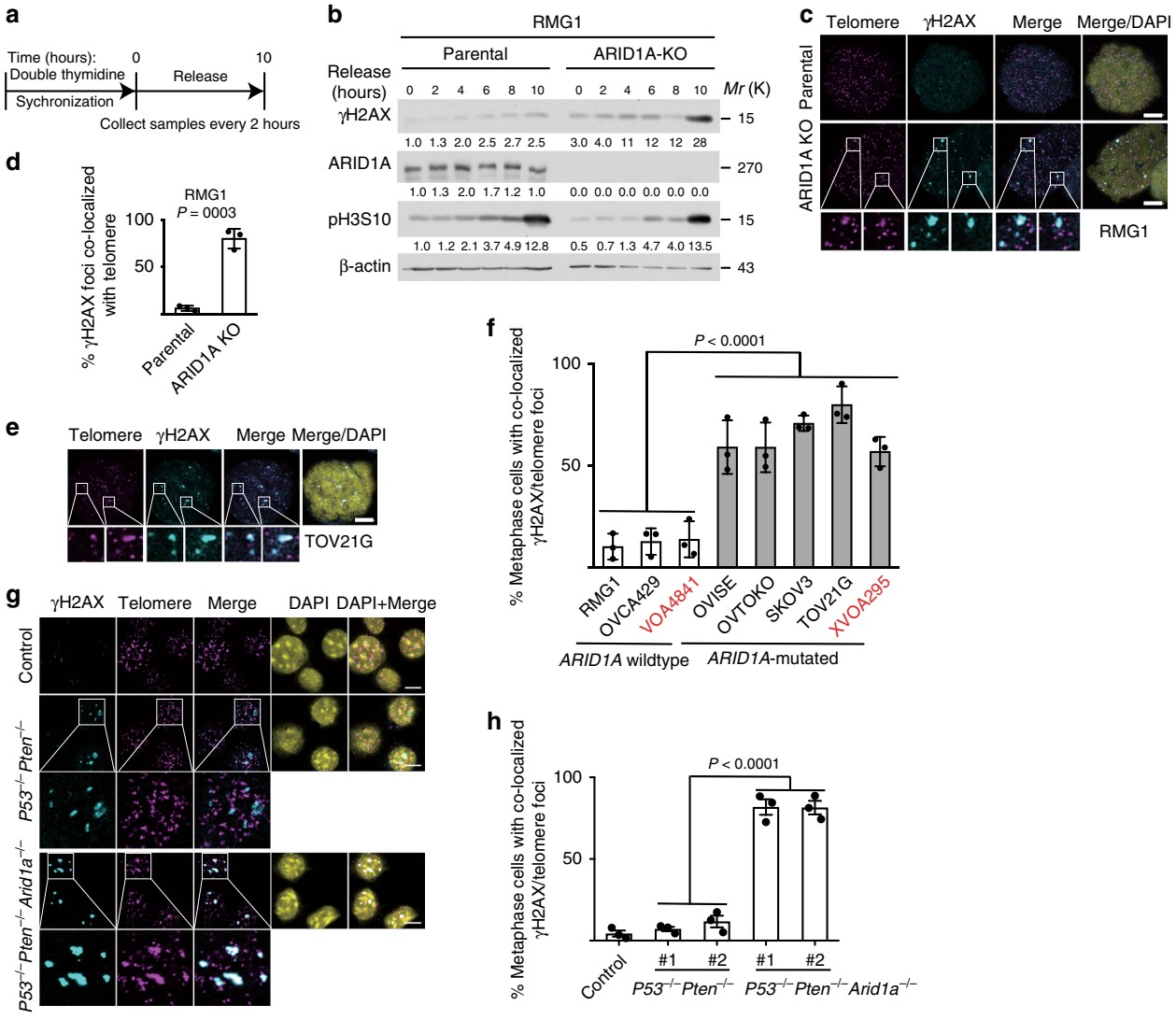

**Fig. 2** ARID1A inactivation causes DNA damage at telomeres. **a, b** Schematic of synchronization and release (**a**) and immunoblot of DNA damage marker γH2AX (**b**) in parental and ARID1A knockout RMG1 cells. Phosphorylated Histone H3 at serine 10 (pH3S10) was used as a marker of mitosis. Relative intensities of immunoblot bands were quantified underneath. **c, d** Co-staining of telomere by FISH and γH2AX (**c**) and quantification of telomeric DNA damage (**d**) in mitotic parental and *ARID1A* knockout RMG1 cells after cytospin. **e, f** Co-staining of telomere by FISH and γH2AX in *ARID1A*-mutated TOV21G mitotic cells (**e**) and quantification of mitotic telomeric DNA damage in a panel of clear cell ovarian cancer cell lines or primary cultures highlighted in red (**f**). **g, h** Representative images (**g**) and quantification (**h**) of telomere DNA damage in parental, $P53^{-/-}/Pten^{-/-}$ and $P53^{-/-}/Pten^{-/-}/Arid1a^{-/-}$ mouse bladder organoid cultures. $n = 3$ independent experiments unless otherwise stated. Data represent mean ± s.e.m. Scale bar = 10 μm. *P* values were calculated using a two-tailed *t* test except in 2f and 2h by multilevel mixed-effects models

inactivation because STAG1 is specifically required for telomere cohesion[10–12]. We validated the association of ARID1A with the *STAG1* promoter by ChIP analysis in *ARID1A* wild-type cells (Fig. 4b) and downregulation of STAG1 at both the mRNA and protein levels in ARID1A KO or knockdown RMG1 and OVCA429 cells (Fig. 4c, d and Supplementary Fig. 4b–e). As a control, ARID1A KO did not decrease expression of the other subunits of cohesin complex, such as STAG2, SMC1, and SMC3 (Fig. 4d and Supplementary Fig. 4f). Notably, loss of ARID1A from the *STAG1* promoter correlated with an increase in the association of ARID1B, the mutually exclusive subunit of mammalian BAF complex[6] (Supplementary Fig. 4g), while core BAF subunit SNF5's association with the *STAG1* promoter was not altered by ARID1A KO (Supplementary Fig. 4h). This suggests that ARID1B is not sufficient to compensate for ARID1A loss in promoting STAG1 expression. Indeed, ARID1B knockdown did

not affect STAG1 expression in RMG1 cells regardless of ARID1A status (Supplementary Fig. 4i). Likewise, STAG1 expression was not altered by SNF5 restoration in SNF5 deficient G401 rhabdoid cells (Supplementary Fig. 4j). This is consistent with the finding that ARID1A loss did not affect SNF5's association with the *STAG1* promoter (Supplementary Fig. 4h). In addition, ARID1A knockdown in nontransformed primary human lung fibroblasts IMR90 cells reduced STAG1 expression, increased telomere damage and anaphase bridges (Supplementary Fig. 4k–n). Notably, compared with *ARID1A* wild-type cells, STAG1 was expressed at significantly lower levels in *ARID1A*-mutated cells at both mRNA and protein levels (Fig. 4e, f). Likewise, STAG1 expression was decreased by ARID1A KO in mouse bladder organoid cultures (Fig. 4g). Finally, in a tissue microarray consisting of 40 cases of OCCCs, expression of ARID1A positively correlated with STAG1 as determined by

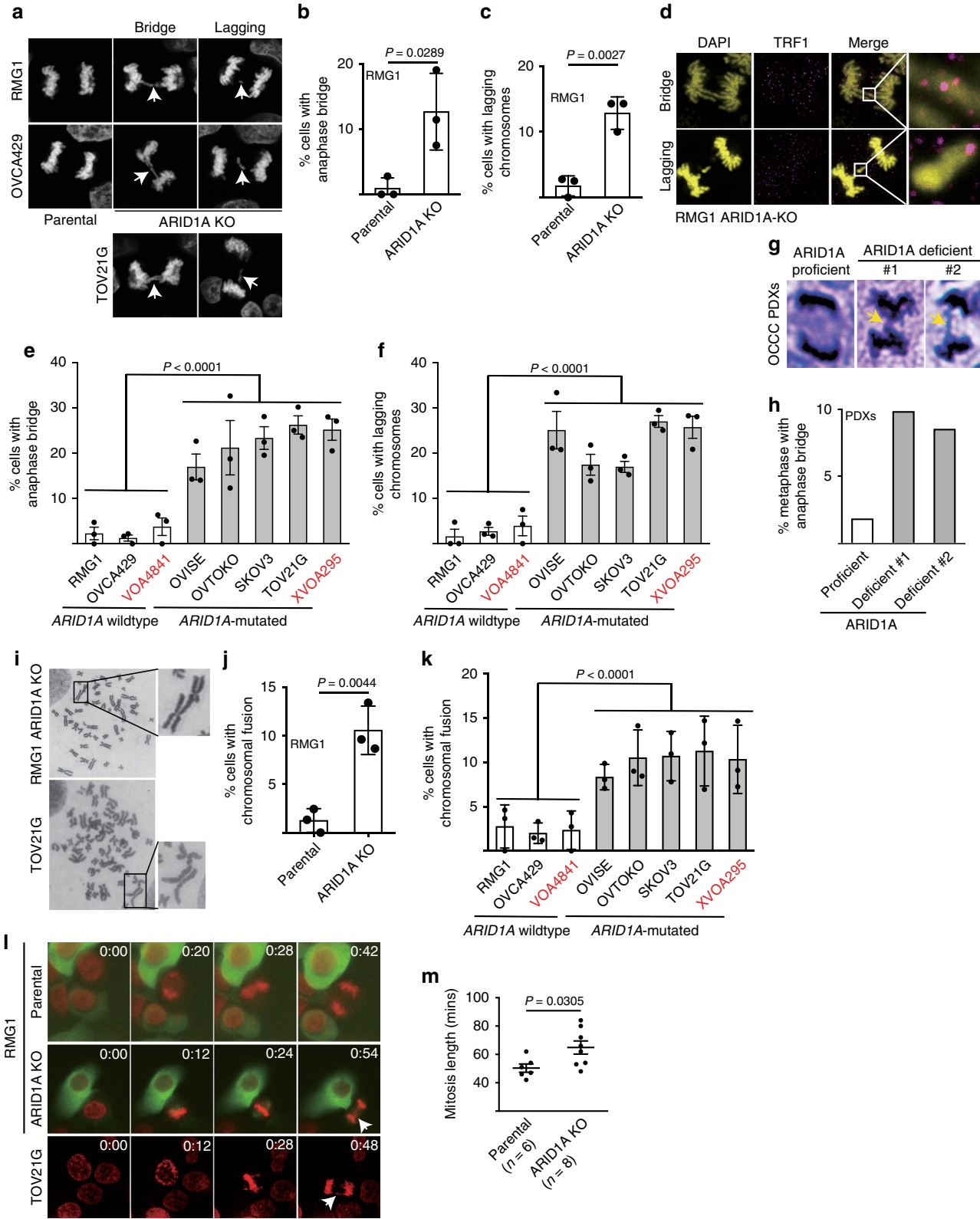

immunohistochemical staining (Fig. 4h). Together, these results support the notion that STAG1 is a direct target of ARID1A-mediated gene transcriptional activation.

**Ectopic STAG1 rescues the defective telomere cohesion.** We next determined whether STAG1 downregulation phenocopies the defects in telomere cohesion observed in ARID1A-inactivated cells. Consistent with previous reports[11,12], STAG1 knockdown in *ARID1A* wild-type RMG1 cells caused an increase in metaphase γH2AX foci, TIFs and an increase in the distance between sister chromatid distal arms (Fig. 5a–d and Supplementary Fig. 5a, b). We next determined whether ectopic STAG1 expression is sufficient to rescue the observed telomere cohesion defects in

**Fig. 3** ARID1A inactivation causes chromosomal defects during mitosis. **a–c** Representative images (**a**) and quantification of anaphase bridge (**b**) and lagging chromosomes (**c**) in parental and ARID1A knockout RMG1 cells. **d** Telomere-binding TRF1 protein staining in anaphase bridges and lagging chromosomes observed in ARID1A knockout RMG1 cells. **e, f** Quantification of percentage of anaphase bridge (**e**) and lagging chromosome (**f**) positive cells in a panel of clear cell ovarian cancer cell lines or primary cultures highlighted in red. **g, h** Representative images of metaphase with anaphase bridge (**g**) and quantification (**h**) in ARID1A proficient and deficient patient-derived xenografts of clear cell ovarian cancer. **i** Representative images of mitotic chromosomal fusion in RMG1 ARID1A knockout and ARID1A-mutated TOV21G cell. **j** Quantification of mitotic chromosomal fusion in parental and ARID1A knockout RMG1 cells. **k** Quantification of mitotic chromosomal fusion in the indicated clear cell ovarian cancer cell lines or primary cultures highlighted in red. **l** Parental and ARID1A knockout RMG1 cells, and ARID1A-mutated TOV21G cells were subjected to time-lapse video microscopic analysis for mitosis. Cell nuclei were visualized by staining for DNA using siR-DNA. Time is expressed as hours: minutes. Arrows points to chromosomal bridges or lagging chromosomes. **m** Quantification of mitosis duration in the indicated cells. $n = 3$ independent experiments unless otherwise stated. Data represent mean ± s. e.m. $P$ values were calculated using a two-tailed $t$ test except for 3e, 3f, and 3k by multilevel mixed-effects models

ARID1A KO cells. To do so, we ectopically expressed a green fluorescent protein (GFP)-tagged wild-type STAG1 in ARID1A KO RMG1 or OVCA429 cells (Fig. 5e–g and Supplementary Fig. 5c, d). As a negative control, we ectopically expressed a mutant STAG1 that lacks a nuclear localization sequence (Fig. 5e–g and Supplementary Fig. 5c, d)[15]. Indeed, ectopically expressed wild-type STAG1, but not the nuclear exclusion mutant, rescued the observed defects in telomere cohesion of sister chromatids, anaphase bridge and lagging chromosomes during mitosis (Fig. 5h–j and Supplementary Fig. 5e, f), and the increase in γH2AX foci and TIFs caused by ARID1A inactivation (Fig. 5k, l and Supplementary Fig. 5g–j). Consistently, ectopically expressed wild-type STAG1, but not the mutant STAG1 rescued the mitotic defects observed using live cell imaging (Fig. 5m and Supplementary Movies 4–6). Together, these results indicate that STAG1 downregulation mediates the defects in telomere cohesion observed in ARID1A-inactivated cells.

**ARID1A inactivation is selective against mitotic cells**. We next sought to determine the fate of cells with or without ARID1A. To do so, we first flow cytometry sorted $G_1$ and $G_2/M$ phase parental and ARID1A KO RMG1 cells based on Hoechst staining. We performed single-cell colony formation assay by the sorted cells. Compared with parental cells, ARID1A KO did not significantly affect the colony formation ability of $G_1$ phase single cells (Fig. 6a, b). In contrast, the colony formation ability of $G_2/M$ phase cells was significantly decreased by ARID1A inactivation (Fig. 6a, b). In addition, we synchronized parental and ARID1A KO RMG1 cells into $G_1$ phase and $G_2/M$ phase of the cell cycle (Supplementary Fig. 6a, b) and obtained similar results using synchronized $G_1$ and $G_2/M$ phase single cells (Fig. 6c, d). Similar observations were made using additional OCCC cell lines and primary cultures (Supplementary Fig. 6b–d). These findings suggest that ARID1A-inactivated cells with severe telomere cohesion defects were selectively eliminated during mitosis. Indeed, ectopic expression of wild-type STAG1 that rescued telomere cohesion defects also rescued the decrease in colony formation ability of $G_2/M$ ARID1A KO single cells (Fig. 6e, f). In contrast, a mutant STAG1 that was unable to rescue the telomere cohesion defects also failed to rescue the decrease in colony formation ability of $G_2/M$ phase ARID1A KO single cells (Fig. 6e, f). Together, our data suggest that ARID1A-inactivated cells with greater genomic instability are lost during $G_2/M$ phase.

Notably, markers of apoptosis such as cleaved caspase 3 and cleaved PARP p85 were expressed at higher levels in ARID1A deficient compared with proficient OCCC cell lines (Fig. 6g). Notably, the observed apoptosis in ARID1A-mutated OCCC cell lines can be suppressed by a pan-Caspase inhibitor zVAD-FMK (Supplementary Fig. 6e). Similar findings were also made in a panel of endometrial carcinoma cell lines (Fig. 6h). Consistently, ARID1A KO significantly decreased the size of tumors formed by RMG1 cells in vivo in an orthotopic xenograft OCCC model

(Fig. 6i, j). In addition, although colony formation ability of $G_1$ phase cells were comparable between ARID1A wild-type and KO cells, the cell growth as indicated by integrated intensity in colony formation assay was significantly decreased by ARID1A KO (Fig. 6k). Similar observations were made using additional OCCC cell lines and primary cultures (Supplementary Fig. 6f). Furthermore, the decrease in the growth of $G_1$ phase ARID1A KO cells can be rescued by a wild-type STAG1, but not by a mutant STAG1 that failed to rescue the telomere cohesion defects (Fig. 6l).

These findings also suggest that ARID1A-mutated cells select against gross chromosomal aberrations through a mechanism involving loss of telomere cohesion followed by apoptosis. Consequently, this selection process enriches for cancer cells lacking genomic instability and thus preserves genomic stability. Indeed, compared with parental controls, mitosis targeting agent such as paclitaxel was less effective in inducing apoptosis in ARID1A KO cells (Supplementary Fig. 6g). This is consistent with the report in the literature that ARID1A expression levels inversely correlates with response to mitosis targeting agent such as paclitaxel in triple-negative breast cancers[16]. We next directly tested this possibility by comparing copy number variations in ARID1A wild-type and mutated uterine corpus endometrial carcinoma, stomach adenocarcinoma, and colon adenocarcinoma in the TCGA dataset because these cancer types display high-ARID1A mutation frequencies[1] and the TCGA does not have OCCCs. Indeed, compared with ARID1A wild-type tumors, ARID1A-mutated tumors displayed significantly less genomic instability as measured by CNA in all tested cancer types (Fig. 6m). Notably, in PTEN mutant and TP53 wild-type uterine corpus endometrial carcinomas, ARID1A-mutated tumors also displayed significantly less CNA compared with ARID1A wild-type tumors (Supplementary Fig. 6h). Together, we conclude that ARID1A inactivation causes selection against the survival of cells with severe defects in telomere cohesion, which correlates with a preservation of genomic stability in ARID1A-mutated cancers.

## Discussion

Here, we show that ARID1A plays a critical role in telomere cohesion and ARID1A inactivation serves as a negative selection to preserve genomic stability. Consistently, in uterine corpus endometrial carcinoma, stomach adenocarcinoma and colon adenocarcinoma, compared with ARID1A wild-type tumors, ARID1A-mutated tumors displayed a significantly lower levels of genomic instability as determined by CNA. Notably, although it was not linked to ARID1A mutational status, previous studies indicate the presence of two distinct clusters of OCCC based on their copy number changes[17]. Regardless, our findings support the notion that ARID1A exerts its tumor suppressor function by preserving genomic stability through eliminating cells with severe genomic instability during mitosis when it is inactivated. Indeed, ARID1A-mutated cells display a higher level of basal apoptosis in vitro and ARID1A KO reduces tumor growth in vivo in an

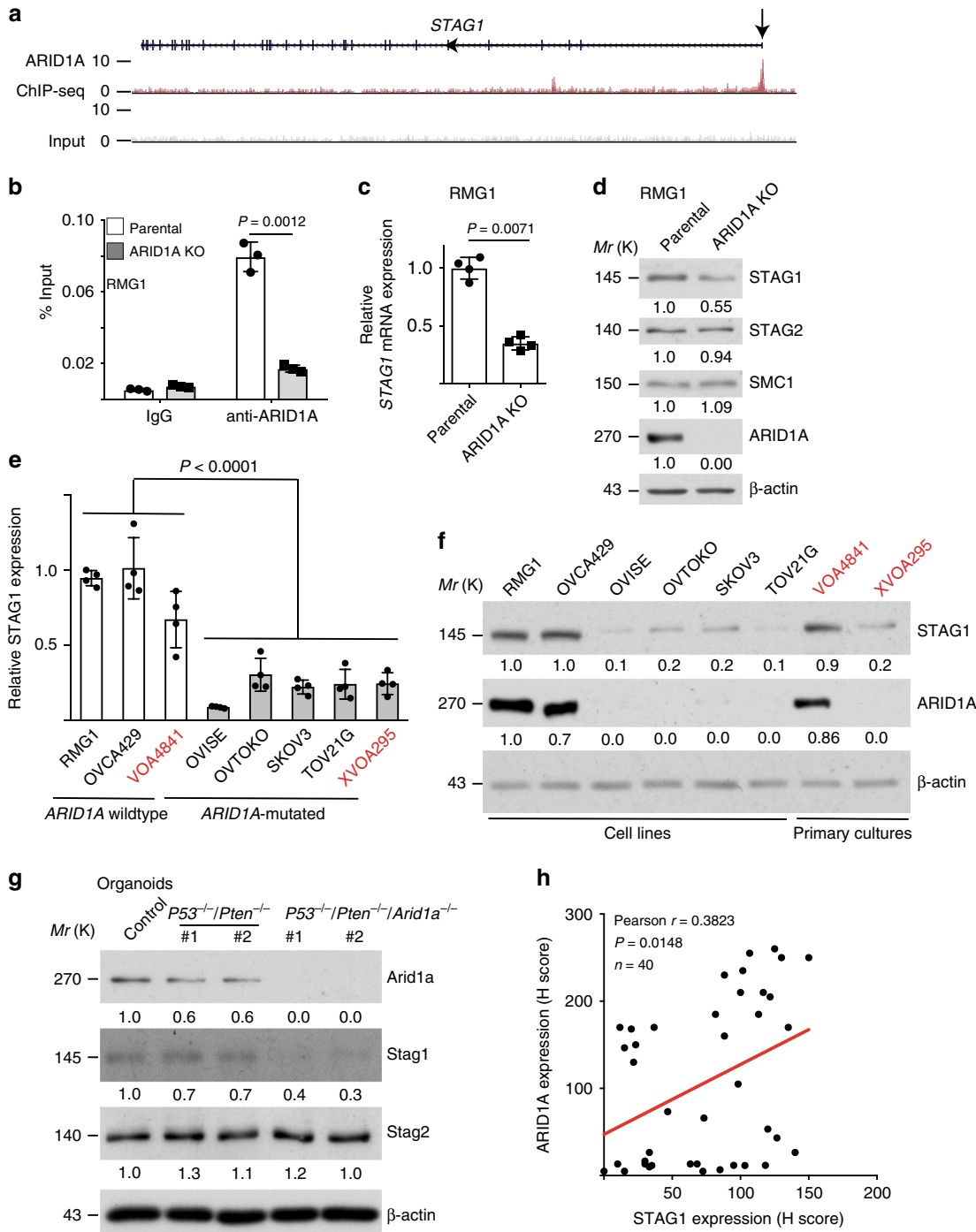

**Fig. 4** ARID1A promotes STAG1 expression. **a** ARID1A ChIP-seq and input tracks of the *STAG1* gene locus in RMG1 cells. **b** Validation of ARID1A binding to the STAG1 promoter by ChIP-qPCR in parental and ARID1A knockout RMG1 cells. **c**, **d** Validation of STAG1 downregulation by ARID1A knockout at the mRNA levels determined by qRT-PCR analysis (**c**) and at the protein levels determined by immunoblot (**d**) in RMG1 cells. Expression of other cohesin subunits STAG2 and SMC1 was used negative controls. **e**, **f** Expression of STAG1 at the mRNA levels determined by qRT-PCR analysis (**e**) and at the protein levels by immunoblot (**f**) in the indicated clear cell ovarian cancer cell lines or primary cultures highlighted in red. **g** Immunoblot of cohesin subunits Stag1 and Stag2 in wild-type controls, *P53⁻/⁻/Pten⁻/⁻* and *P53⁻/⁻/Pten⁻/⁻/Arid1a⁻/⁻* mouse bladder organoid cultures. **h** Correlation analysis between the expression of ARID1A and STAG1 in tumor microarray (TMA) of clear cell ovarian carcinomas determined by immunohistochemical staining. *n* = 3 independent experiments unless otherwise stated. Data represent mean ± s.e.m. *P* values were calculated using a two-tailed *t* test except for 4e by multilevel mixed-effects models, and for 4 h by Pearson correlation analysis. Relative intensities of immunoblot bands were quantified underneath

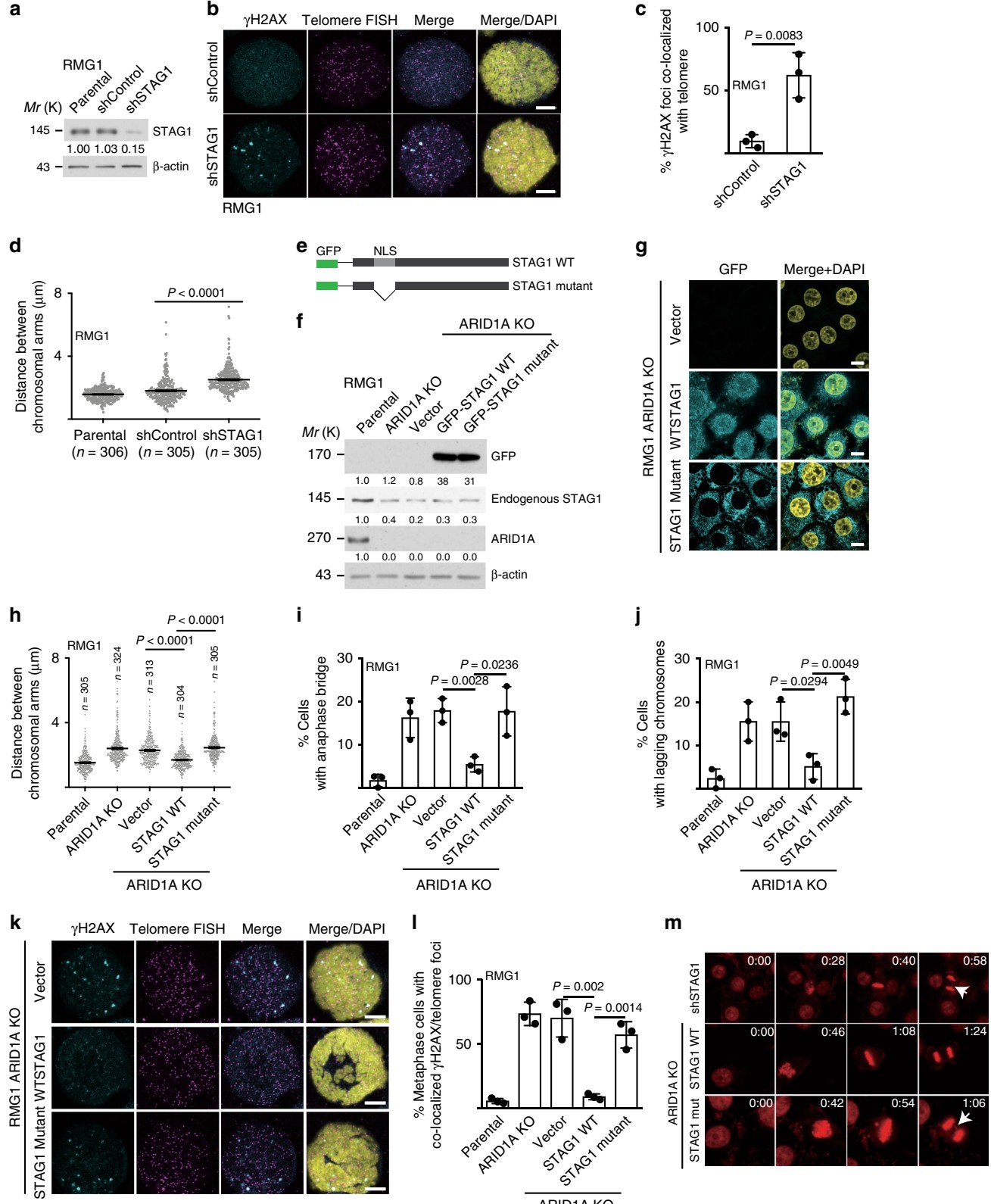

orthotopic xenograft model. Consistently, ARID1A inactivation prolongs survival in an *Apc-* and *Pten*-defective mouse ovarian cancer model[18]. Cells with a complete lack of cohesion cannot proliferate and genomic instability is tumor promoting. Interestingly, ARID1A functions to resolve this paradox by reducing telomere cohesion to allow for elimination of cells with severe genomic instability during mitosis when it is inactivated. Similar to *ARID1A* mutation, although mutations in cohesin subunits causes chromosomal abnormalities and aneuploidy in models systems such as mouse embryonic fibroblasts[11,12], cancers associated with mutations in cohesin subunits are often not associated with aneuploidy and genomic instability[19,20].

**Fig. 5** Ectopic STAG1 rescues the telomere damage and mitotic defects in ARID1A-inactivated cells. **a** Immunoblot validation of STAG1 knockdown in RMG1 cells. **b, c** Representative images (**b**) and quantification (**c**) of telomere DNA damage in RMG1 shRNA vector control and STAG1 knockdown cells determined by telomere FISH and γH2AX co-staining. **d** Quantification of distance between distal ends of sister chromatids enriched by colcemid treatment from the indicated RMG1 cells. **e–g** Schematics of STAG1 wild-type and mutant that lacks nuclear localization sequence (**e**), and validation of ectopic STAG1 expression by immunoblot (**f**) or immunofluorescence (**g**) in *ARID1A* knockout RMG1 cells. **h–j** Quantification of distance between distal ends of sister chromatids (**h**), and percentage of anaphase bridge (**i**) and lagging chromosome (**j**) positive-mitotic cells in the indicated parental, *ARID1A* knockout, and *ARID1A* knockout RMG1 cells rescued with wild-type or mutant STAG1, respectively. **k, l** Co-staining of telomere FISH and γH2AX (**k**) and quantification of telomeric DNA damage (**l**) in mitotic parental, *ARID1A* knockout, and *ARID1A* knockout RMG1 cells rescued with wild-type or mutant STAG1. **m** RMG1 cells expressing shSTAG1 or *ARID1A* knockout RMG1 cells rescued with wild-type or mutant STAG1 were subjected to time-lapse video microscopic analysis. Cell nuclei were visualized by staining for DNA using siR-DNA. Time is expressed as minutes: seconds. Arrows point to examples of lagging chromosomes. *n* = 3 independent experiments unless otherwise stated. Data represent mean ± s.e.m. Scale bar = 10 μm. *P* values were calculated using a two-tailed *t* test. Relative intensities of immunoblot bands were quantified underneath

Our data are consistent with the idea that $G_1$ sorted cells have already been selected through the $G_2/M$ phase and, thus, only those cells with chromosome stability and integrity survived comparing with those going through the $G_2/M$ phase. This explains why compared with $G_2/M$-sorted cells, $G_1$-sorted cells demonstrated a better percentage of colony formation capability on a single cell basis for ARID1A-inactivated cells (e.g., Fig. 6b, d and Supplementary Fig. 6c). However, $G_1$-sorted single cells have to cycle through mitosis to replicate in order to form a colony. Indeed, compared with wild-type cells, colonies formed by ARID1A-inactivated cells were significantly lower in integrated intensity, a marker for cell growth (e.g., Fig. 6k). Together, these results support a continuous selection process during the subsequent division of $G_1$ phase ARID1A-inactivated cells.

In addition to cohesin, sister chromatids catenation contributes to cohesion that interlocks DNA between newly replicated sister chromatids[21,22]. Notably, cohesin hinders decatenation by TOP2[22] and thus a reduction in cohesin may facilitate decatenation by TOP2. This suggests that a decrease in cohesin may help cells overcome the reduction of chromatin-associated TOP2A induced by ARID1A inactivation to allow for proliferation of *ARID1A*-mutated cells[7]. Our results clearly demonstrated a critical role of STAG1 in the observed phenotypes because ectopic STAG1 is sufficient to rescue the telomere cohesion defects. A limitation of our study is that other factors including TOP2A may also contribute to the process. In addition, TOP2A defects caused by ARID1A inactivation creates an increased reliance on ATR checkpoint and inhibition of ATR triggers premature mitotic entry, genomic instability and apoptosis[8]. This suggests that ATR may participate in the negative selection against *ARID1A*-mutated cells during mitosis. Indeed, we show that DNA damage marker γH2AX foci formation is specifically localized to telomere. Consistently, it has been shown that cohesin promotes restart of replication forks at difficult to replicate regions such as telomeres[12,23,24] and ATR plays a critical role in restarting of replication forks[25].

In summary, our results show that ARID1A plays a critical role in telomere cohesion by promoting STAG1 expression. The defective telomere cohesion is selective against genomic instability caused by ARID1A inactivation during mitosis to balance the need to proliferation and the tumor suppressive function of ARID1A in preserving genomic stability. Thus, our study provides mechanistic understanding of the long-standing paradox between ARID1A's role in maintaining mitotic integrity and the lack of genomic instability in *ARID1A*-mutated cancers.

## Methods

**Cell lines**. IMR90 human diploid fibroblasts were cultured according to American Type Culture Collection (ATCC) under low-oxygen tension (2%) in Dulbecco's Modified Eagle medium (DMEM, 4.5 g/L glucose) supplemented with 10% fetal bovine serum (FBS), L-glutamine, sodium pyruvate, nonessential amino acids, and sodium bicarbonate. All experiments were performed on IMR90 fibroblasts

between population doublings #25 and 35. Primary human ovarian clear cell cultures were published previously[26]. The protocol for using primary cultures of human ovarian clear cell tumor cells was approved by the University of British Columbia Institutional Review Board. Informed consent was obtained from human subjects. All relevant ethical regulations have been complied with. The primary tumor cells were cultured in RPMI 1640 supplemented with 10% FBS and 1% penicillin/streptomycin. The culture of clear cell ovarian cancer cell lines including RMG1, OVCA429, OVISE, OVTOKO, SKOV3, and TOV21G was performed as we previously described[27,28]. G401 rhabdoid tumor cell line was purchased from ATCC and cultured in McCoy's 5a modified medium supplemented with 10% FBS and 1% penicillin/streptomycin. Endometrial cancer cell lines ARK1, ARK2, SPAC1L, SPAC1S, AN3CA, RL95, HEC1A, Ishikawa, SNG-M, EN1, SPEC2, and SNG-II were provided by Dr. Vijayalakshmi Shridhar. SPAC1L, SPAC1S, and SPEC2 were cultured in RPMI 1640 supplemented with 10% FBS and 1% penicillin/streptomycin. The rest cell lines were cultured in DMEM/F12 supplemented with 10% FBS and 1% penicillin/streptomycin. All the cells lines are authenticated at The Wistar Institute's Genomics Facility using short tandem repeat DNA profiling. Regular mycoplasma testing was performed using the LookOut Mycoplasma polymerase chain reaction (PCR) detection (Sigma). Each of the experiments was performed in duplicate in three independent experimental repeats.

**Three-dimensional bladder organoids culture**. Mouse bladder organoids (wide type, $P53^{f/f}$, $Pten^{f/f}$, $P53^{f/f}$, $Pten^{f/f}$; $Arid1a^{f/f}$) were generated by Drs. Lijie Rong and Cory T. Abate-Shen[29–31]. Briefly, $p53^{f/f}$; $Pten^{f/f}$ mice[31] were crossed with $Arid1a^{f/f}$ mice (Jackson Laboratory, Jax no. 027717) to generate $p53^{f/f}$; $Pten^{f/f}$; $Arid1a^{f/f}$ mice. The $p53^{f/f}$; $Pten^{f/f}$ and $p53^{f/f}$; $Pten^{f/f}$; $Arid1a^{f/f}$ mice were further crossed with $R26$-$CAG$-$EYFP$ mice (Jackson Laboratory, Jax no. 007903) to trace bladder tumor cells by the activation of EYFP reporter via Cre-mediated gene recombination. Tumor induction was achieved by injection of an adenovirus expressing Cre-recombinase (adeno-Cre, University of Iowa Vector Core Facility) into the bladder lumen (ref1). Wild-type bladder organoid was generated from normal bladder urothelium of noninduced $p53^{+/+}$; $Pten^{+/+}$; $R26$-$CAG$-$EYFP$ mice. Bladder tumor or normal bladder tissue was dissociated with collagenase/hyaluronidase (STEMCELL Technologies #07912)[29]. Fluorescence activated cell sorting for either YFP+ cells (tumor tissue) or cells expressing either EpCAM or E-cadherin (normal tissue)[30]. Sorted cells were plated in 96-well low-attachment plate (Corning #3474) to generate organoids. Both organoids were maintained in the hepatocyte medium (Corning #355056) plus 5% Matrigel, 10 ng/mL EGF (Corning #3555056), 5% heat-inactivated, charcoal-stripped FBS (Gibco #12676), 1× Glutamax (Gibco #35050), 10 μM ROCK inhibitor Y-27632 (STEMCELL Technologies #07171) and 1× antibiotic-antimycotic (Thermo-Fisher #15240-062). Organoids were passaged weekly by 0.25% trypsin digestion. Single cells were collected for cytospin and telomere FISH and γH2AX co-staining analysis.

**Reagents and antibodies**. Thymidine was purchased from Sigma (Cat. no. T1895). Colcemid was purchased from ThermoFisher (Cat. no. 15212012). Hoechst 33342 was purchased from ThermoFisher (Cat no. 62249). Propidium Iodide (PI) was purchased from ThermoFisher (Cat. no. P1304MP). zVAD-FMK was purchased from Santa Cruz (Cat. no. sc-3067). The following antibodies were obtained from the indicated suppliers: mouse anti-γH2AX (Millipore, Cat. no. 05-636, 1:1000 for immunoblotting and 1:500 for immunofluorescence), rabbit anti-ARID1A (Cell signaling technology, Cat. no. 12354S, 1:1000 for immunoblotting, 1:1000 for immunohistochemical (IHC), 5 μg/IP for ChIP), mouse anti-ARID1B (Abgent, Cat. no. AT1190a, 1:1000 for immunoblotting, and 5 μg/IP for ChIP), rabbit anti-SNF5 (Bethyl Laboratories, Cat. no. A301-087, 1:1000 for immunoblotting, 5 μg/IP for ChIP), rabbit anti-STAG1 (Bethyl Laboratories, Cat. no. A302–579A, 1:1000 for immunoblotting), mouse anti-STAG2 (Novus Biologicals, Cat. no. MAB16661, 1:1000 for immunoblotting), rabbit anti-SMC1 (Bethyl Laboratories, Cat. no. A300-055A, 1:1000 for immunoblotting), rabbit anti-phospho-Histone H3 (Ser10) (Millipore, Cat. no. 06-570, 1:1000 for immunoblotting), mouse anti-β-actin (Sigma, Cat. no. A2228, 1:10,000 for immunoblotting), anti-centromere antibodies protein (derived from human CREST patient

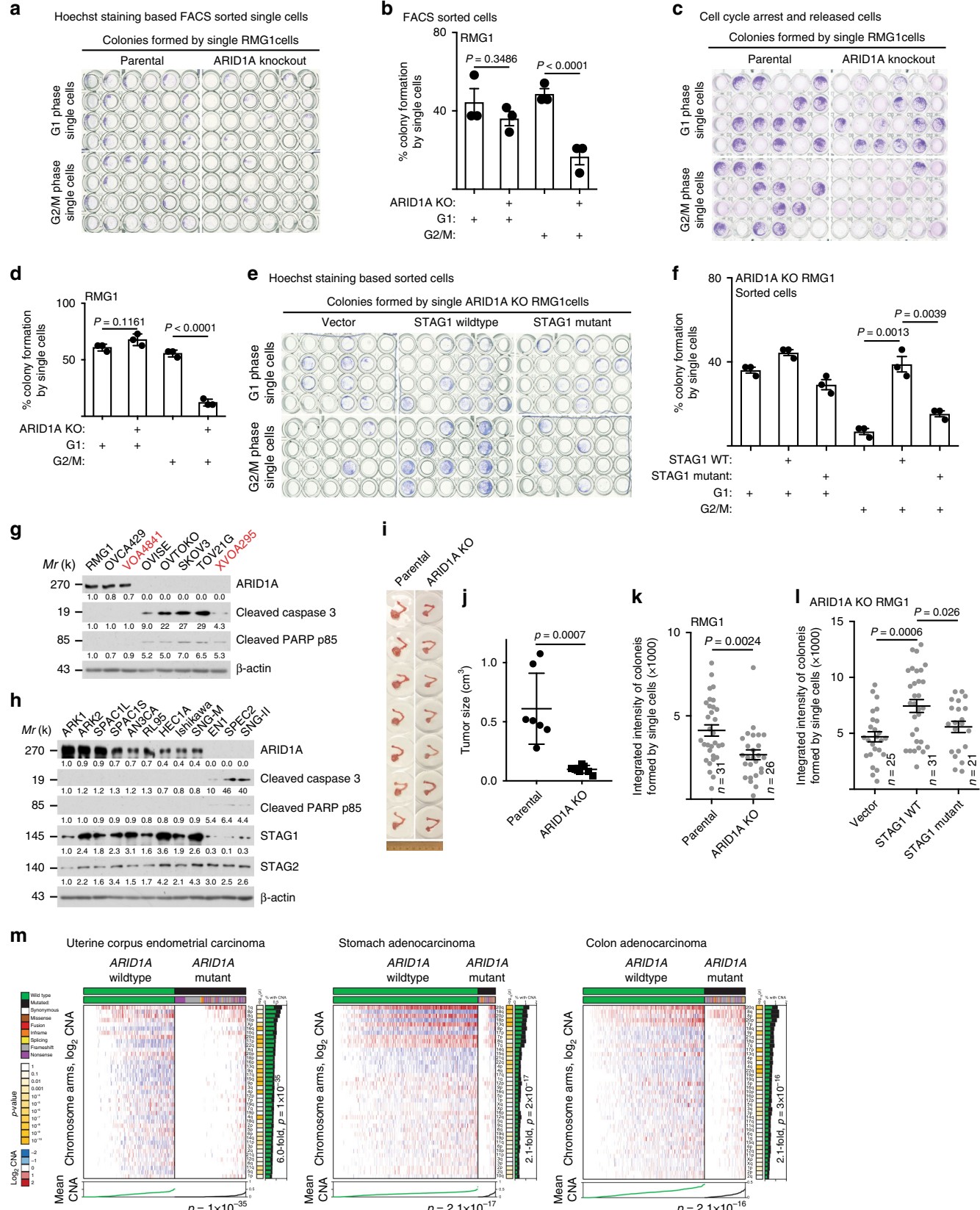

serum, Antibodies, Inc., Cat. no. 15-235-0001, 1:200 for immunofluorescence), mouse anti-FLAG tag (Sigma, Cat. no. F1804, 1:1000 for immunoblotting), rabbit anti-GFP tag (ThermoFisher, Cat. no. MA5-15256, 1:1000 for immunoblotting, and 1:500 for immunofluorescence), rabbit anti-cleaved PARP p85 (Promega, Cat. no. G7341, 1:1000 for immunoblotting), rabbit anti-cleaved caspase 3 (Cell Signaling, Cat. No: 9661, 1:1000 for immunoblotting), mouse anti-SMC3 (Santa Cruz, Cat. No: sc-376352, 1:1000 for immunoblotting). The secondary antibodies used were raised against mouse or rabbit and conjugated with Alexa 488 (ThermoFisher, Cat. no. A-10680) or Alexa 555 (ThermoFisher, Cat. no. A-21428).

**Fig. 6** ARID1A inactivation is selective against the survival of cells during mitosis. **a**, **b** Representative images (**a**) and quantification of colony formation efficiency (**b**) of colonies formed by single parental or *ARID1A* knockout RMG1 cells at the indicated G$_1$ or G$_2$/M phases of the cell cycle sorted by flow cytometry based on Hoechst 33342 staining. **c**, **d** Representative images (**c**) and quantification of colony formation efficiency (**d**) of colonies formed by single parental or *ARID1A* knockout RMG1 cells at the indicated synchronized G$_1$ or G$_2$/M phases of the cell cycle. **e**, **f** Representative images (**e**) and quantification of colony formation efficiency (**f**) of colonies formed by single *ARID1A* knockout RMG1 cells rescued with wild-type or mutant STAG1 at the indicated G$_1$ or G$_2$/M phases of the cell cycle sorted by flow cytometry based on Hoechst 33342 staining. **g**, **h** Expression of ARID1A and apoptosis markers cleaved caspase 3 or cleaved PAPR p85 in a panel of clear cell ovarian cancer cell lines (**g**) or endometrial cancer cell lines (**h**), respectively. **i**, **j** Images of orthotopic tumors formed by parental and *ARID1A* knockout RMG1 cells (**i**) and the sizes of the tumors formed were quantified (**j**). **k**, **l** Integrated density analysis of colonies formed by single cell G$_1$ phase RMG1 parental and *ARID1A* knockout cells (**k**) or *ARID1A* knockout RMG1 cells rescued by wild-type or mutant STAG1 (**l**). **m** Compared with *ARID1A* wild-type tumors, *ARID1A*-mutated tumors exhibit a significant less copy number variations in the indicated cancer types in the TCGA datasets. *n* = 3 independent experiments unless otherwise stated. Data represent mean ± s.e.m. *P* values were calculated using a two-tailed *t* test except in 6 m by multilevel mixed-effects models. Relative intensities of immunoblot bands were quantified underneath

**In vivo animal model**. PDX models were established by direct implantation of surgically removed human ovarian clear cell tumor tissues orthotopically in the bursal sac of the immunocompromised mice under a protocol approved by the Institutional Animal Care and Use Committee of the Wistar Institute. Tumor tissue procurement was approved by the Institutionally Review Board at Christiana Care Health System and the Wistar Institute. The protocols are approved by the Institutional Animal Care and Use Committee of the Wistar Institute. For orthotopic xenograft ovarian cancer model[32], 1 × 10$^6$ RMG1 parental and *ARID1A* KO RMG1 cells were unilaterally injected into the ovarian bursa sac of 6- to 8-week-old female NSG mice. After four weeks, tumors were surgically dissected and tumor size was calculated as 1/2 (length × width$^2$).

For *Arid1a*$^{-/-}$/*Pik3ca*$^{H1047R}$ genetic clear cell ovarian tumor mouse model, the transgenic mice were generated by crossing *Arid1a*$^{flox/flox}$ mice with *R26*-*Pikca*$^{H1047R}$ (Jackson Laboratory, Jax no. 016977)[26]. All mice were maintained in specific pathogen-free barrier facilities. Administration of intrabursal adeno-Cre was used to induce OCCC after adeno-Cre injection[32]. After 5 weeks, mice were euthanized and tumors were surgically dissected. Tumors were cut into small pieces and digested with 0.25% trypsin/PBS to get single tumor cells. The single cells were used for analysis.

**Synchronization by double thymidine treatment**. For cell synchronization[33], cells were treated first with 2 mM thymidine for 18 h, followed by 9 h release under normal cultural conditions, then treated again with 2 mM thymidine for 18 h. Cells were washed with prewarmed PBS and incubated in prewarmed fresh medium. Cells were subsequently collected at 0, 2, 4, 6, 8h, and 10 h for cell cycle analysis, or subjected to western blot analysis. Single cells from G$_1$ phase or G$_2$/M phase were used for single cell colony formation assay.

**Telomere-γH2AX immuno-FISH**. Cells were collected by shake-off after double thymidine synchronization to G$_2$/M phase, and swollen in 0.075 M KCl hypotonic buffer for 10 min at 37 °C. The cells were fixed by 1% formaldehyde in PBS for 2 min, and then spun onto coverslips using a cytospin apparatus (Cytospin). Chromosome spreads were fixed again in 4% formaldehyde in PBS for 15 min, followed by permeabilization in 0.5% Triton X-100/PBS for 15 min at room temperature. For the telomere PNA-γH2AX immuno-FISH[34], the mitotic cells on slides were incubated with TAMRA-OO-[CCCTAA]3 labeled PNA probe (PANAGENE, Cat. no. F2001) at 85 °C for 2 min, then incubated in 37 °C overnight. After formamide fixation, the cells were stained with γH2AX antibody. DAPI counter staining was performed to label the nuclei or chromosome. Stained slides were analyzed using a Leica TCS SP5 II scanning confocal microscope.

**Prometaphase chromosome spread analysis**. Mitotic cells were collected either by shake-off or by colcemid (50 ng/mL) enrichment for 3 h[11,35]. Cells were incubated in 0.075 M KCl followed by an overnight fixation in methanol/acetic acid (3:1). Chromosome spreads were generated by dropping cells onto −80 °C pre-cooled glass slides. Slides were next stained in Giemsa staining solution (Sigma) for 4 min. Stained slides were analyzed for sister chromosome separation by Nikon Eclipse 80i microscope[11]. Spreads without Giemsa staining were used for telomere FISH. Telomere FISH was performed as detailed above using TAMRA-OO-[CCCTAA]3 labeled PNA probe (PANAGENE, Cat. no. F2001)[36].

**Live cell time-lapse microscopy imaging**. For live cell time-lapse microscopy imaging[37], RMG1 cells were plated into glass bottom 6-well plate with CellLight Tubulin-GFP, BacMam 2.0 (Thermo Fisher Scientific) to visualize the microtubules and incubated overnight. To visualize the nuclei, SiR-DNA regent (Cytoskeleton Inc., Cat. n o. CY-SC007) was added in the medium right before filming. Time-lapse fluorescence and DIC video microscopy were performed for 24 h with Nikon Te300 inverted microscope (20× objective). Images were acquired by using NIS Elements AR software.

**Constructs and lentivirus infection**. Constructs of GFP-tagged STAG1 and GFP-tagged STAG1ΔNLS were kindly provided by Dr. Anna Kurlandzka[15]. GFP-STAG1 and GFP-STAG1ΔNLS were cloned into the pLVX lentivirus vector and validated by sequencing. pLKO.1-shARID1A (TRCN0000059090), pLKO.1-shARID1B (TRCN0000107361) and pLKO.1-shSTAG1 (TRCN0000144850) were obtained from Molecular Screening Facility at Wistar Institute. pLKO.1-shRNA and pLVX system were used for lentivirus package. HEK293FT cell was transfected by Lipofectamine 2000. Lentivirus was harvested and filtered with 0.45 μm filter 48 hours post transfection. Cells infected with lentiviruses were selected in 1 μg/ml puromycin 48 h post infection.

**Immunoblotting**. Cells were lysed in 1× sample buffer (2% sodium dodecyl sulphate (SDS), 10% glycerol, 0.01% bromophenol blue, 62.5 mM Tris, pH 6.8, and 0.1 M DTT) and heated to 95 °C for 10 min. Protein concentrations were determined using the protein assay dye (Bio-Rad, Cat. No: #5000006) and Nanodrop. An equal amount of total protein was resolved using SDS polyacrylamide gel electrophoresis gels and transferred to PVDF membranes at 110 V for 2 h at 4 °C. Membranes were blocked with 5% nonfat milk in TBS containing 0.1% Tween 20 (TBS-T) for 1 h at room temperature. Membranes were incubated overnight at 4 °C in the primary antibodies in 4% BSA/TBS + 0.025% sodium azide. Membranes were washed four times in TBS-T for 5 min at room temperature, after which they were incubated with Horseradish peroxidase-conjugated secondary antibodies (Cell Signaling Technology) for 1 h at room temperature. After washing four times in TBS-T for 5 min at room temperature, proteins were visualized on film after incubation with SuperSignal West Pico PLUS Chemiluminescent Substrate (Thermo Fisher Scientific). Unprocessed images of scanned immunoblots shown in Figures and Supplementary Figures are provided in a Source Data file.

**Quantification PCR with reverse transcription (RT)**. Total RNA was isolated using Trizol (Invitrogen) according to the manufacturer's instruction. Extracted RNAs were used for reverse-transcriptase PCR (RT-PCR) with High-Capacity cDNA Reverse Transcription Kit (Thermo Fisher). Quantitative PCR (qPCR) was performed using QuantStudio 3 Real-Time PCR System. The primers sequences used for quantitative RT-PCR are as follows: *STAG1* forward: 5′-GCCTACT TGGTGGACAGTTTAT-3′ and reverse: 5′- CCTCTCCTTGAACAGGTTCTTC-3′; *β-microglobulin (B2M)* forward: 5′-GGCATTCCTGAAGCTGACA-3′ and reverse: 5′-CTTCAATGTCGGATGGATGAAAC-3′. B2M was used as an internal control.

**Single-cell colony formation assay**. Single cells were collected either by double thymidine synchronization to G$_1$ or G$_2$/M phase, or by direct flow cytometry sorting based on Hoechst 33342 staining. Single cells were picked up by using mouth pipette and long Pasteur glass pipette[38]. Single cell was seeded into one well of 96-well plate. Cells were cultured for three weeks and crystal violet staining was performed to visualize the colonies formed by the single cells[32].

**ChIP and quantification PCR**. Cells were cross-linked with 1% formaldehyde for 10 min at room temperature. The reaction was quenched by 0.125 M glycine for 5 min. Fixed cells were lysated with ChIP lysis buffer 1 (50 mM HEPES-KOH (pH 7.5), 140 mM NaCl, 1 mM EDTA (pH 8.0), 1% Triton X-100, and 0.1% DOC) on ice and lysis buffer 2 (10 mM Tris (pH 8.0), 200 mM NaCl, 1 mM EDTA, and 0.5 mM EGTA) at room temperature. Chromatin was digested with MNase in digestion buffer (10 mM Tris 8.0, 1 mM CaCl$_2$, and 0.2% Triton X-100) at 37 °C for 15 min. The nucleus was broken down by one pulse of bioruptor with high output. Chromatin was incubated overnight at 4 °C and protein A + G Dynabeads were added to the reaction for another 1.5 h. Magnetic beads were washed and chromatin was eluted and reversed. Chromatin was then treated with proteinase K and purified with Gel extraction kit (Qiagen, cat. no. 28706). ChIP DNA was used for

**ChIP-qPCR.** For ChIP-qPCR, the following primers of the *STAG1* gene transcriptional start site (TSS) region were used: forward: 5′-CCCTGCTCCTA CTTGGATTTAG-3′ and reverse: 5′-TCACTCTTGCCTGGTGAAAG-3′.

**Tumor microarray analysis.** Tumor microarray of clear cell ovarian carcinoma was constructed and provided by Dr. Ronny Drapkin. For IHC staining[26], antigens were unmasked using citrate buffer (Thermo Fisher, Cat. no. 005000). Endogenous peroxidases were quenched with 3% hydrogen peroxide in methanol. Staining was performed by using an antibody against ARID1A (Abcam, Cat. no. ab182560, 1:500 dilution) or an antibody against SATG1 (Bethyl Laboratories, Cat. no. A302-579A, 1:500 dilution) on consecutive sections.

**Telomere length assay.** For telomere length assay[39], genomic DNA was isolated using genomic DNA purification kit (Promega) and digested with AluI and MboI. Equal amounts of digested DNA (~4 μg) were separated by 0.7% agarose gel electrophoresis in 1× TBE, denatured, and transferred to a GeneScreen Plus membrane (PerkinElmer). The blot was crosslinked, hybridized at 42 °C with 5′-end-labeled $^{32}$P-(TTAGGG)$_4$ probe in Church buffer, and washed twice for 5 min each with 0.2 M wash buffer (0.2 M Na$_2$HPO4 pH 7.2, 1 mM EDTA, and 2% SDS) at room temperature and once for 10 min with 0.1 M wash buffer at 42 °C. The images were analyzed by Phosphor-imager, visualized by Typhoon 9410 Imager (GE Healthcare), and processed with ImageQuant 5.2 software (Molecular Dynamics).

**Bioinformatic analysis.** Putative direct downstream targets of ARID1A responsible for chromosome organization and segregation were identified using data from GSE120060 GEO dataset[14], specifically samples for RMG1 ARID1A CHIP-seq (GEO id GSM3392689), CHIP input (GEO id GSM3392698) and RMG1 ARID1A wild-type (WT) and ARID1A KO RNA-seq (GEO ids GSM3392681, GSM3392682, and GSM3392683, GSM3392684). ARID1A peaks were identified using HOMER algorithm[40] and peaks that passed FDR < 5% threshold and had signal at least fivefold over input were considered significant. Genes with significant ARID1A peaks at the transcription start site that were found to be significantly downregulated by ARID1A KO (DESeq2 algorithm[41], FDR < 1%, at least 1.2-fold) were identified as putative direct ARID1A targets (645 coding genes). Genes from gene ontology group GO:0007059—chromosome segregation were considered for candidate selection.

TCGA ARID1A mutation status and overall CNA for chromosome arm level was downloaded from Firebrowse. Log$_2$ CNA values for each arm as well as mean absolute log$_2$ CNA levels across all arms were compared between *ARID1A* wild-type and mutated samples using unpaired *t* test. Significance of difference of percent of samples with nonzero CNA at each arm between mutated and wild-type ARID1A was estimated using paired *t* test.

**Statistical analysis and reproducibility.** Statistical analysis was performed using GraphPad Prism 7 (GraphPad) for Mac OS or Stata MP 15 (StataCorp LP, 4905 Lakeway Drive, College Station, TX 77845, USA. www.stat.com). *t* test was used for comparison of means between two groups. Multilevel mixed-effects model was applied to determine the difference of the studied outcome (e.g., distance between distal chromosomal arms) between *ARID1A* wild-type and mutated cell lines. Experiments were repeated three times unless otherwise stated. The representative images were shown unless otherwise stated. Quantitative data are expressed as mean ± s.e.m. unless otherwise stated. Imaging analysis was performed blindly without known the experimental groups but not randomly based on each of the different experimental groups were examined together.

## Data availability

Previously published ChIP-seq and RNA-seq data for ARID1A wild-type and ARID1A knockout RMG1 cells are available at the Gene Expression Ominibus (GEO) under access code GSE120060[14]. For correlation between ARID1A mutational status and copy number variations, the TCGA uterine corpus endometrial carcinoma [https://doi.org/10.7908/C16M36B7], stomach adenocarcinoma [https://doi.org/10.7908/C1D50MFS] and colon adenocarcinoma [https://doi.org/10.7908/C1M61JMS] datasets were used. Other datasets referenced during the study are available from cBioPortal [https://www.cbioportal.org/] and Firebrowse websites [http://firebrowse.org/]. All the other data supporting the findings of this study are available within the article and its supplementary information files and from the corresponding author upon reasonable request. A reporting summary for this article is available as a Supplementary Information file. The source data underlying Figs. 1e, g, h, 2b, d, f, h, 3b, c, e, f, h, j–m, 4b–g, 5a, f, l, 6b, d, f, g, h, j, k, l, and Supplementary Figs. 1a, b, 2b, d, f, m, 3a, b, c, 4b–k, m, n, 5b, 5d–f, 5h–j, 6c–g are provided as a Source Data file.

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

## Acknowledgements

We thank Drs. David Huntsman and Yeming Wang for primary ovarian clear cell carcinoma cultures, Dr. Jose Conejo-Garica for the genetic OCCC model, Dr. Anna Kurlandzka for the GFP-tagged STAG1 and GFP-tagged STAG1ΔNLS constructs, Dr. Vijayalakshmi Shridhar for endometrial cancer cell lines and Dr. S Hua for technical assistance. This work was supported by US National Institutes of Health grants (R01CA160331, R01CA163377, R01CA202919 and R01CA239128 to R.Z., P01AG031862 to R.Z., P50CA228991 to R.Z., R01CA140652 to P.M.L., P01CA221757 to C.T.A.-S., R50CA211199 to A.V.K.), US Department of Defense (OC150446 and OC180109 to R.Z.), The Honorable Tina Brozman Foundation for Ovarian Cancer Research (to R.Z.) and Ovarian Cancer Research Alliance (Collaborative Research Development Grant to R.Z., and Ann and Sol Schreiber Mentored Investigator Award to S.W.). Support of Core Facilities was provided by Cancer Center Support Grant (CCSG) CA010815 to The Wistar Institute.

## Author contributions

B.Z., J.L., L.R., Z.D., S.W., N.F., J.Z., T.F., and A.V.K. performed the experiments and analyzed data. B.Z., C.T. A.-S., and R.Z. designed the experiments. Q.L. performed the statistical analysis. S.J., M.G.C., M.E.B., and R.D. contributed key experimental materials. P.M.L., C.T.A-S., and R.Z. supervised the studies. B.Z., L.R., Z.D., P.M.L., R.Z. wrote the paper. R.Z. conceived the study.

## Additional information

**Competing interests:** The authors declare no competing interests.

