## [Peer Review File · Nature Communications]

Reviewers' comments:

Reviewer #1 (Remarks to the Author):

In this manuscript Zhao et al., report that loss of the BAF complex subunit ARID1A results in a loss of telomere cohesion, which in turn maintains genetic stability in ARID1A mutant cancers by selectively eliminating gross chromosomal aberrations during cancer. Overall the manuscript is extensive, thorough, with high quality data and will certainly be of interest to the wider field. Below I address a few minor points that should be addressed to strengthen/clarify the paper prior to publication:

- All western blots would benefit from quantitation.

- In Figure 2 the authors look at co-staining of telomeres via FISH and γ H2AX via IF. This data would benefit from additional staining to confirm that the cytospun cells are indeed in metaphase (H3 S10P), instead of simply relying on the DAPI stain.

- In Fig S2i the authors perform an important control, showing that γ H2AX foci do not overlap with a centromeric FISH probe, suggesting the damage is confined to telomeres. The authors only include a representative image for this. As this is important to their conclusion this data will require the same analysis and quantification as has been performed for the telomeric probe.

- In Figure 3 it would be insightful to know if these are specifically anaphase bridges originating from the telomeres?

- Fig. S4a requires more detail. A key should be included for expression level colours. Why are there two columns shown for the expression levels of both the WT and KO cells? Are these biological replicates?

- By cross-referencing RNA-seq data with ARID1A ChIP-seq data the authors determine that the telomere specific cohesion subunit STAG1 is a direct target of ARID1A, the expression of which is decreased upon loss of ARID1A. The authors go on to demonstrate that this is reflected at the protein level by Immunoblotting. The authors state on line 147 "As a control, ARID1A knockout did not decrease the expression of the other subunits of cohesion complex such as STAG2 and SMC1 (Fig. 4d)." Notably, the authors do not include the SMC3 subunit in this Western blot analysis. Of note, the authors show in Fig. S4a that SMC3 expression levels are reduced upon ARID1A KO but do not comment on, or address this further.

- In Figure 6 a-e it is not clear why no defect in colony formation is observed in the G1 sorted cells, as these single cells are presumably also having to cycle through mitosis to replicate and form a colony?

Reviewer #2 (Remarks to the Author):

In this manuscript, the authors investigate two apparently paradoxical observations; first that ARID1A plays a key role in decatenating newly replicated sister chromatids, a requirement for proper chromosome segregation during mitosis, and second that there appear to be few copy number alterations in ARID1A-mutated cancers.

In a series of experiments that predominantly utilise established and primary ovarian clear cell carcinoma (OCCC) cell lines, the authors show that ARID1A loss in ARID1A-wild-type cells does indeed induce chromosomal segregation abnormalities during mitosis, which are mediated by loss of the cohesin component STAG1. However, the authors suggest that cells with ARID1A- mutations are selectively lost during mitosis, potentially via activation of apoptosis. Thus, they hypothesise that chromosomally abnormal cells are lost, leaving behind cells with relatively stable genomes resulting in tumours with few copy number aberrations.

Major points

1. All the discovery experiments are performed in established OCCC cell lines that are ARID1A-wildtype in which ARID1A is either knocked out or knocked down. There are some comparisons between wildtype and mutant cell lines, but there are no rescue experiments performed in the mutant cells – what happens with ARID1A is re-expressed in, for example, OVTOKO cells?
2. Most ARID1A-mutant OCCC also harbours mutation in PIK3CA. Thus, losing ARID1A function appears to require aberrant PI3K signalling to maintain viability/tumorigenesis. What happens to chromosomal segregation and CN phenotypes when both these genes are mutated?
3. If ARID1A loss activates apoptosis (Figure 6g, 6h), treatment with zVAD.fmk should reverse the observed phenotypes. Is this observed?
4. The observation that single cells in G2/M are unable to form colonies whilst those in G1 are is interesting but curious. As the authors note, those single cells must pass through G2/M successfully in order to form colonies. What are the kinetics of mitosis in these different cell populations? Is mitosis prolonged? At what point are these cells dying? And why do the surviving

cells survive? What allows them to escape apoptosis? Does this survival influence their susceptibility to agents that target mitosis (e.g. taxanes) These are utterly critical questions to explain the formation of ARID1A-mutated cancers.

5. Figure 6m – analysis of TCGA endometrial cancer data. Whilst it is superficially informative to investigate the whole TCGA endometrial cancer cohort, delineation by ARID1A status alone is too simplistic – the serous (TP53 mutated) cluster largely lacks ARID1A mutations and does indeed has very high levels of CNA. However, this is driven by loss of p53 and this cohort is completely distinct and will distort the analysis. Similarly, the POLE-mutated and MSI cohorts are also distinct, with their own driver mutations. Thus, this analysis should be restricted to the endometrioid cohort that are nearly all PTEN mutant/TP53 wild-type: this will answer the question of whether ARID1A loss (seen in c.40%) is truly associated with CN stability.

6. Tan et al performed CN analysis on a cohort of 50 OCCC cases (Tan et al Clin. Cancer Res. 2011 17:1521) and demonstrated two distinct patterns of CNA. I do not believe that ARID1A analysis (IHC or sequencing) was performed, but this manuscript should at least be included in the discussion.

7. Finally, a partially philosophical point. Are established ARID1A-wildtype OCCC cell lines the correct model to use? Evidence indicates no difference in clinical behaviour or outcome between ARID1A-wildtype and mutant OCCC, suggesting that there may be other events in wild-type cells that phenocopy ARID1A loss. Moreover, the wildtype cells are derived from tumours that have formed despite normal ARID1A function – a more appropriate model to study ARID1A function would be ES cells (as used by Dykhuizen et al) or some other non-transformed model.

Minor points

1. As the authors will be aware (Domke et al Nat. Commun 2013), there is considerable controversy as to the nature of SKOV3 cells. In addition, TOV21G appears to be hypermutated, and has functional abnormalities in ARID1A, ARID1B, PIK3CA, PTEN and RAS amongst many others that are likely to compound results here.

2. Many of the results rely upon scoring of chromosomal events. The methods section contains the expression “Imaging analysis was performed blindly but not randomly.” What does this mean? How many people scored images? And if the analyses were not random, how were cells/nuclei selected for quantification?

3. Stats, using figure 5h and 5i as examples. The methods state that t-tests were employed. However, in these cases, ANOVA with multiple comparisons testing should be employed.

4. In figure 2d for example, is ‘% cells with co-localized gH2AX/telomere foci’ the correct analysis? A cell with a single co-localised focus scores the same as a cell with 100 co-localised foci. Surely the correct analysis should be ‘number of co-localised foci/nucleus’?

Point-by-Point Response to The Reviewers Comments

We sincerely thank both the Reviewers and the Editor for the constructive and thoughtful review provided for our manuscript. All the comments raised are truly valuable to improve the manuscript. Correspondingly, we have strived to provide new experimental answers to their comments. I hope that there is no doubt that we have taken the Reviewers' comments very seriously. We believe that by addressing the reviewers' concerns we have produced a more solid and cohesive manuscript. A point-by-point response to the reviewers' comments is detailed below with original comments italicized. Changes that directly address the reviewers' concerns were denoted with vertical lines in the right margin in the revised manuscript. We hope the Reviewers and the Editors will find this revised manuscript to be suitable for publication.

Reviewers' comments:

Reviewer #1 (Remarks to the Author):

In this manuscript Zhao et al., report that loss of the BAF complex subunit ARID1A results in a loss of telomere cohesion, which in turn maintains genetic stability in ARID1A mutant cancers by selectively eliminating gross chromosomal aberrations during cancer. Overall the manuscript is extensive, thorough, with high quality data and will certainly be of interest to the wider field. Below I address a few minor points that should be addressed to strengthen/clarify the paper prior to publication:

- All western blots would benefit from quantitation.

Response: As requested, all western blots have now been quantified.

- In Figure 2 the authors look at co-staining of telomeres via FISH and γ H2AX via IF. This data would benefit from additional staining to confirm that the cytopun cells are indeed in metaphase (H3 S10P), instead of simply relying on the DAPI stain.

Response: We thank the reviewer for the comment. As requested, we now performed the suggested experiments and confirmed that the cytopun cells are indeed in metaphase as indicated by ~100% positive for H3 S10P staining (**new data Supplementary Fig. 2g-h**)

- In Fig S2i the authors perform an important control, showing that γ H2AX foci do not overlap with a centromeric FISH probe, suggesting the damage is confined to telomeres. The authors only include a representative image for this. As this is important to their conclusion this data will require the same analysis and quantification as has been performed for the telomeric probe.

Response: We thank the reviewer for the comment. As requested, we now performed the suggested experiments and quantified the results the same as the analysis we performed for the telomeric probe. The results show that ARID1A knockout did not significantly increase γ H2AX foci that co-localized with centromere (**new data Supplementary Fig. 2I**).

- In Figure 3 it would be insightful to know if these are specifically anaphase bridges originating from the telomeres?

Response: We agree the reviewer and now performed the telomere-associated TRF1 staining in the anaphase bridges. Indeed, our results show that the anaphase bridges are positive for TRF1 staining, suggesting that they were originated from the telomeres (**new data Fig. 3d**).

- Fig. S4a requires more detail. A key should be included for expression level colours. Why are there two columns shown for the expression levels of both the WT and KO cells? Are these biological replicates?

Response: We thank the reviewer for spotting this, and apologize for the confusion and omission. As requested, we now added a color key to the Figure and clarified that the two columns represent biological replicates.

- By cross-referencing RNA-seq data with ARID1A ChIP-seq data the authors determine that the telomere specific cohesion subunit STAG1 is a direct target of ARID1A, the expression of which is decreased upon loss of ARID1A. The authors go on to demonstrate that this is reflected at the protein level by Immunoblotting. The authors state on line 147 "As a control, ARID1A knockout did not decrease the expression of the other subunits of cohesion complex such as STAG2 and SMC1 (Fig. 4d)." Notably, the authors do not include the SMC3 subunit in this Western blot analysis. Of note, the authors show in Fig. S4a that SMC3 expression levels are reduced upon ARID1A KO but do not comment on, or address this further.

Response: We thank the reviewer for pointing this out. As requested, we now evaluated the SMC3 protein expression in parental and ARID1A KO cells by immunoblot. Our results show that similar to STAG2 and SMC1, the expression of SMC3 was not decreased by ARID1A knockout (**new data Supplementary Fig. 4f**). Notably, since SMC3 is a core cohesin subunit and thus is required for both telomere and centromere cohesion. Therefore, our result that SMC3 expression was not changed by ARID1A knockout is consistent with our findings that telomere but not centromere cohesion was impaired in ARID1A inactivated cells.

- In Figure 6 a-e it is not clear why no defect in colony formation is observed in the G1 sorted cells, as these single cells are presumably also having to cycle through mitosis to replicate and form a colony?

Response: We thank the reviewer for the comment. Our data are consistent with the idea that G1-sorted cells have already been selected through the G2/M phase, thus only those cells with chromosome stability and integrity survived comparing with those going through the G2/M phase. This explains why compared with G2/M-sorted cells, G1-sorted cells demonstrated a better percentage of colony formation capability for ARID1A-inactivated cells (e.g., **Fig. 6b, 6d and Supplementary Fig. 6c**). However, as rightly pointed out the reviewer, G1-sorted single cells have to cycle through mitosis to replicate and form a colony. Indeed, as predicated by the reviewer, compared with wild-type cells, colonies formed by ARID1A knockout cells were significantly lower in integrated intensity, a marker for cell growth (e.g., **Fig. 6k**). Together, these results support a continuous selection process during the subsequent division of G1 sorted ARID1A inactivated cells, which explains the decrease in the growth of colonies formed by G1-sorted single ARID1A knockout cells compared with those formed by G1 sorted single parental wild-type cells. We now included these discussions on page 12, paragraph 2.

Reviewer #2 (Remarks to the Author):

In this manuscript, the authors investigate two apparently paradoxical observations; first that ARID1A plays a key role in decatenating newly replicated sister chromatids, a requirement for proper chromosome segregation during mitosis, and second that there appear to be few copy number alterations in ARID1A-mutated cancers.

In a series of experiments that predominantly utilise established and primary ovarian clear cell carcinoma (OCCC) cell lines, the authors show that ARID1A loss in ARID1A-wild-type cells does indeed induce chromosomal segregation abnormalities during mitosis, which are mediated by loss of the cohesin component STAG1. However, the authors suggest that cells with ARID1A^{-/-} mutations are selectively lost during mitosis, potentially via activation of apoptosis. Thus, they hypothesise that chromosomally abnormal cells are lost, leaving behind cells with relatively stable genomes resulting in tumours with few copy number aberrations.

Major points

1. All the discovery experiments are performed in established OCCC cell lines that are ARID1A-wildtype in which ARID1A is either knocked out or knocked down. There are some comparisons between wildtype and mutant cell lines, but there are no rescue experiments performed in the mutant cells – what happens with ARID1A is re-expressed in, for example, OVTOKO cells?

Response: We thank the reviewer for the comment. As requested, we now performed the suggested rescue experiments in OVTOKO cells by re-expressing ARID1A. Our new results show that ARID1A re-expression suppressed DNA damage at telomeres and reduced percentage of cells with anaphase bridge (**new data Supplementary Fig. 2m-p**).

2. Most ARID1A-mutant OCCC also harbours mutation in PIK3CA. Thus, losing ARID1A function appears to require aberrant PI3K signalling to maintain viability/tumorigenesis. What happens to chromosomal segregation and CN phenotypes when both these genes are mutated?

Response: The reviewer is correct that most ARID1A-mutant OCCC cells often harbour mutation in PIK3CA. However, our results showed that similar results were obtained from both cells with *ARID1A* and *PIK3CA* mutations (such as TOV21G), and *ARID1A* mutated but *PIK3CA* wildtype (such as OVISE and OVTOKO) (e.g., Fig. 1e, 2f, 3e, 3f, 3k and 6g) (1,2). Thus, the observed changes occurred regardless of *PIK3CA* mutational status.

3. If ARID1A loss activates apoptosis (Figure 6g, 6h), treatment with zVAD.fmk should reverse the observed phenotypes. Is this observed?

Response: We thank the reviewer for the suggestion. As requested, we now performed the suggested experiments and showed that zVAD-FMK treatment substantially reversed the observed apoptosis phenotypes (**new data Supplementary Fig. 6e**).

4. The observation that single cells in G2/M are unable to form colonies whilst those in G1 are is interesting but curious. As the authors note, those single cells must pass through G2/M successfully in order to form colonies. What are the kinetics of mitosis in these different cell populations? Is mitosis prolonged? At what point are these cells dying? And why do the surviving cells survive? What allows them to escape apoptosis? Does this survival influence their susceptibility to agents that target mitosis (e.g. taxanes) These are utterly critical questions to explain the formation of ARID1A-mutated cancers.

Response: We thank the reviewer for the comment. As requested, we now performed the suggested analysis and showed that the mitosis was prolonged in these cells (**new data Fig. 3m**). This is consistent with other reports in the literature (3). Our results suggest that this serves as a selection mechanism to eliminate the severely damaged cells during G2/M because we showed that G2/M phase single cells showed a reduced colony formation capability. However, not all the G2/M cells will die and those with chromosome stability will survive, which explains the observed reduction in colony formation instead of a complete elimination of colony formation capability. Thus, the process increases genomic stability of the surviving cells by eliminating the ones with chromosomal instability. Consistently, compared with *ARID1A* wild-type tumors, *ARID1A*-mutated tumors displayed significantly less genomic instability as measured by copy number alterations. Indeed, compared with parental wild-type controls,

mitosis targeting agent such as paclitaxel was less effective in inducing apoptosis in ARID1A knockout cells (**Fig. 1 for reviewer**). This is consistent with the report in the literature that ARID1A expression levels inversely correlates with response to mitosis targeting agent such as paclitaxel in triple-negative

breast cancers (4).

5. Figure 6m – analysis of TCGA endometrial cancer data. Whilst it is superficially informative to investigate the whole TCGA endometrial cancer cohort, delineation by ARID1A status alone is too simplistic – the serous (TP53 mutated) cluster largely lacks ARID1A mutations and does

indeed has very high levels of CNA. However, this is driven by loss of p53 and this cohort is completely distinct and will distort the analysis. Similarly, the POLE-mutated and MSI cohorts are also distinct, with their own driver mutations. Thus, this analysis should be restricted to the endometrioid cohort that are nearly all PTEN mutant/TP53 wild-type: this will answer the question of whether ARID1A loss (seen in c.40%) is truly associated with CN stability.

Response: We thank the reviewer for the insightful comment. As requested, we now performed the suggested analysis and our new results show that in endometrioid cohort with PTEN mutant/TP53 wild-type, ARID1A mutated tumors displayed significantly less copy number alterations compared with ARID1A wild-type tumors (**new data Supplementary Fig. 6g**).

6. Tan et al performed CN analysis on a cohort of 50 OCCC cases (Tan et al Clin. Cancer Res. 2011 17:1521) and demonstrated two distinct patterns of CNA. I do not believe that ARID1A analysis (IHC or sequencing) was performed, but this manuscript should at least be included in the discussion.

Response: We thank the reviewer for the suggestion. As requested, we now included the paper in the discussion on page 12, paragraph 1 (Ref. 16).

7. Finally, a partially philosophical point. Are established ARID1A-wildtype OCCC cell lines the correct model to use? Evidence indicates no difference in clinical behaviour or outcome between ARID1A-wildtype and mutant OCCC, suggesting that there may be other events in wild-type cells that phenocopy ARID1A loss. Moreover, the wildtype cells are derived from tumours that have formed despite normal ARID1A function – a more appropriate model to study ARID1A function would be ES cells (as used by Dykhuizen et al) or some other non-transformed model.

Response: We thank the reviewer for the comment. As requested, we now performed the ARID1A knockdown experiments in non-transformed, primary human embryonic lung fibroblast IMR90 cells and show that similar to *ARID1A* wildtype OCCC cells, ARID1A knockdown reduced STAG1 expression, increased telomere damage and anaphase bridge in non-transformed IMR90 cells (**new data Supplementary Fig. 4k-n**)

Minor points

1. As the authors will be aware (Domke et al Nat. Commun 2013), there is considerable controversy as to the nature of SKOV3 cells. In addition, TOV21G appears to be hypermutated, and has functional abnormalities in ARID1A, ARID1B, PIK3CA, PTEN and RAS amongst many others that are likely to compound results here.

Response: We thank the reviewer for the comment. As consistently demonstrated throughout the manuscript, our results were obtained in panel of ARID1A-mutated cell lines and primary cultures that include OVISE, OVTOKO, XVOA295 in addition to TOV21G and SKOV3. Based on genetic profiles of these cell lines (1,2), the only consistent genetic alterations commonly occur among the cell lines is ARID1A mutation.

2. Many of the results rely upon scoring of chromosomal events. The methods section contains the expression “Imaging analysis was performed blindly but not randomly.” What does this mean? How many people scored images? And if the analyses were not random, how were cells/nuclei selected for quantification?

Response: We thank the reviewer for the comment. As requested, we now clarified the description “Imaging analysis was performed blindly without known the experimental groups and

not randomly based on each of the different experimental groups were quantified together” on page 25, paragraph 1. The images were scored by two persons. Cells/nuclei on a given slide were selected randomly for quantification.

3. Stats, using figure 5h and 5i as examples. The methods state that t-tests were employed. However, in these cases, ANOVA with multiple comparisons testing should be employed.

Response: We thank the reviewer for the comment. We consulted with our biostatistician Dr. Qin Liu, a co-author on this manuscript. Because the comparison was limited two groups (vector vs. wildtype or wildtype vs. mutant) instead of across all five different groups, t-tests is appropriate in these cases. For those comparison across the different groups, we used multilevel mixed-effects models (e.g., Fig. 1e and 1h). We further clarified these points in the methods section on page 24, paragraph 2.

4. In figure 2d for example, is ‘% cells with co-localized γ H2AX/telomere foci’ the correct analysis? A cell with a single co-localised focus scores the same as a cell with 100 co-localised foci. Surely the correct analysis should be ‘number of co-localised foci/nucleus’?

Response: We thank the reviewer for the comment. The reviewer is correct that to limit the bias, we counted the percentage of γ H2AX positive foci that co-localized with telomere. Specifically, we examined >100 γ H2AX foci and determined whether there is any co-localization with telomere to limit the potential bias because the control cells have substantially less γ H2AX foci. We repeated the experiments three times biologically. Thus, we totally counted >300 γ H2AX foci in each of the indicated groups. As such, we corrected the Y-axis label accordingly in the revised manuscript.

Cited References

1. Anglesio MS, Wiegand KC, Melnyk N, Chow C, Salamanca C, Prentice LM, *et al.* Type-specific cell line models for type-specific ovarian cancer research. *PLoS one* **2013**;8:e72162
2. Domcke S, Sinha R, Levine DA, Sander C, Schultz N. Evaluating cell lines as tumour models by comparison of genomic profiles. *Nat Commun* **2013**;4:2126
3. Williamson CT, Miller R, Pemberton HN, Jones SE, Campbell J, Konde A, *et al.* ATR inhibitors as a synthetic lethal therapy for tumours deficient in ARID1A. *Nat Commun* **2016**;7:13837
4. Lin YF, Tseng IJ, Kuo CJ, Lin HY, Chiu IJ, Chiu HW. High-level expression of ARID1A predicts a favourable outcome in triple-negative breast cancer patients receiving paclitaxel-based chemotherapy. *J Cell Mol Med* **2018**;22:2458-68

REVIEWERS' COMMENTS:

Reviewer #1 (Remarks to the Author):

The authors have done an excellent job in addressing my comments and I am now happy to recommend the manuscript for publication.

Reviewer #2 (Remarks to the Author):

In this revised manuscript, the authors have undertaken a series of new experiments to address the reviewer comments.

This reviewer's original comments included the following:

1. Re-expression of ARID1A in ARID1A-mutant cells. The authors have now re-expressed ARID1A in OVTOKO cells, and demonstrated that there was reduced DNA damage at telomeres and fewer cells with anaphase bridges
2. Reversal of apoptosis with zVAD.fmk – the authors have indicated that treatment of SKOV3 and TOV21G cells with zVAD reduces caspase-3 and PARP cleavage
3. Formation of colonies in the absence of ARID1A. Both reviewers noted that successful colony formation, by definition, requires cells to pass through G2/M. However, the previous data suggested that cells lacking ARID1A were selectively lost during G2/M. The authors now show slower mitosis. The authors speculate that cells with greater instability are lost, although they do not directly demonstrate this at a single cell/single colony level.
4. Potential sensitivity to paclitaxel. In their reply to this reviewer's comments, the authors show some data suggesting decreased apoptosis in response to two doses of paclitaxel in ARID1A knockout cells compared to isogenic ARID1A WT cells. These data are extremely interesting and important – the authors should consider incorporating these into the main results section.
5. Comparison to TCGA. The authors have now limited their comparison to PTEN mutant/TP53 wild-type tumours, which I believe to be a more relevant comparison – they show that there are still fewer CNA in the ARID1A-mutant compared to the wild-type tumours.
6. Normal cells. The authors show that loss of ARID1A function in normal cells has the same effect as in malignant cells, suggesting that the data are not an artefact of malignant transformation.

Overall, the reviewers have made stringent efforts to address the reviewer comments.

Point-by-Point Response to The Reviewers Comments.

REVIEWERS' COMMENTS:

Reviewer #1 (Remarks to the Author):

The authors have done an excellent job in addressing my comments and I am now happy to recommend the manuscript for publication.

Response: We thank the reviewer for the positive comments and the positive recommendation.

Reviewer #2 (Remarks to the Author):

In this revised manuscript, the authors have undertaken a series of new experiments to address the reviewer comments.

This reviewer's original comments included the following:

1. Re-expression of ARID1A in ARID1A-mutant cells. The authors have now re-expressed ARID1A in OVTOKO cells, and demonstrated that there was reduced DNA damage at telomeres and fewer cells with anaphase bridges.

Response: We thank the reviewer for the positive comments.

2. Reversal of apoptosis with zVAD.fmk – the authors have indicated that treatment of SKOV3 and TOV21G cells with zVAD reduces caspase-3 and PARP cleavage.

Response: We thank the reviewer for the positive comments.

3. Formation of colonies in the absence of ARID1A. Both reviewers noted that successful colony formation, by definition, requires cells to pass through G2/M. However, the previous data suggested that cells lacking ARID1A were selectively lost during G2/M. The authors now show slower mitosis. The authors speculate that cells with greater instability are lost, although they do not directly demonstrate this at a single cell/single colony level.

Response: We thank the reviewer for the comments. Accordingly, we have now explicitly softened the conclusion on page 10, paragraph 1.

4. Potential sensitivity to paclitaxel. In their reply to this reviewer's comments, the authors show some data suggesting decreased apoptosis in response to two doses of paclitaxel in ARID1A knockout cells compared to isogenic ARID1A WT cells. These data are extremely interesting and important – the authors should consider incorporating these into the main results section.

Response: We thank the reviewer for the positive comments. As requested, we have now incorporated these results into the manuscript as Supplementary Fig. 6g.

5. Comparison to TCGA. The authors have now limited their comparison to PTEN mutant/TP53 wild-type tumours, which I believe to be a more relevant comparison – they show that there are still fewer CNA in the ARID1A-mutant compared to the wild-type tumours.

Response: We thank the reviewer for the positive comments.

6. Normal cells. The authors show that loss of ARID1A function in normal cells has the same effect as in malignant cells, suggesting that the data are not an artefact of malignant transformation.

Response: We thank the reviewer for the positive comments.

Overall, the reviewers have made stringent efforts to address the reviewer comments.

Response: We thank the reviewer for the positive comments.